# A Soft Matrix Enhances the Cancer Stem Cell Phenotype of HCC Cells

**DOI:** 10.3390/ijms20112831

**Published:** 2019-06-10

**Authors:** Boren Tian, Qing Luo, Yang Ju, Guanbin Song

**Affiliations:** 1Key Laboratory of Biorheological Science & Technology, Ministry of Education, College of Bioengineering, Chongqing University, Chongqing 400030, China; t_boren@163.com (B.T.); Qing.luo@cqu.edu.cn (Q.L.); 2Department of Mechanical Science and Engineering, Nagoya University, Nagoya 464-8603, Japan; ju@mech.nagoya-u.ac.jp

**Keywords:** hepatocellular carcinoma, matrix stiffness, hydrogel, plasticity, cancer stem cells

## Abstract

Cancer stem cells (CSCs) comprise a small portion of cancer cells, have greater self-renewal ability and metastatic potential than non-CSCs and are resistant to drugs and radiotherapy. CSCs and non-CSCs, which can reversibly change their stemness states, typically play roles in plasticity and cancer cell heterogeneity. Furthermore, the component that plays a key role in affecting CSC plasticity remains unknown. In this study, we utilized mechanically tunable polyacrylamide (PA) hydrogels to simulate different stiffnesses of the liver tissue matrix in various stages. Our results showed that hepatocellular carcinoma (HCC) cells were small and round in a soft matrix. The soft matrix increased the expression levels of liver cancer cells with stemness properties (LCSC) surface markers in HCC cells and the number of side population (SP) cells. Moreover, the soft matrix elicited early cell cycle arrest in the G1 phase and increased the cell sphere-forming ability. In addition, cells grown on the soft matrix showed enhanced chemoresistance and tumorigenicity potential. In summary, our study demonstrated that a soft matrix increases the stemness of HCC cells.

## 1. Introduction

Cancer is the second most serious disease in the world after cardiovascular disease. Hepatocellular carcinoma (HCC) is one of the most common malignancies with a high mortality rate worldwide due to its resistance to chemotherapy and poor prognosis [1]. Liver cancer cells with stemness properties (LCSCs) comprise a small portion of HCC cells. Compared with non-LCSCs, LCSCs have great self-renewal and metastatic potential. Furthermore, LCSCs are insensitive to drugs and radiotherapy and play a key role in the metastasis and recurrence of HCC [2]. Cancer stem cells (CSCs) typically have plasticity, which means that CSCs and non-CSCs can reversibly transform between stem and non-stem cell states. CSC plasticity plays a key role in cancer progression and metastasis [3,4]. Although previous studies have shown the importance of CSC plasticity, the underlying factors remain unknown.

Previous studies have shown that the biomechanical component is crucial in the regulation of stem cell differentiation. Studies have shown that different matrix stiffnesses can induce mesenchymal stem cells (MSCs) into osteoblasts or adipocytes [5]. The biomechanical component, a key element of the tumor microenvironment, has been shown to be an important factor affecting the plasticity of CSCs in recent studies. LCSCs differentiate into cancer cells under fluid shear stress via the Wnt/β-catenin signaling pathway [6]. Additionally, shear stress can increase the CSC phenotype of lung cancer cells [7], and matrix stiffness promotes the development of breast CSCs via modulation of integrin-linked kinase (ILK) [8]. The matrix stiffness also mediates colorectal cancer cell stemness via the Yes-associated protein [9].

Both clinical and in vitro research have indicated that cancerous tissue exhibits mechanical heterogeneity. Research has shown that the stiffnesses of the invasive front and core are different, as the invasive front is stiffer than the core [10,11]. More importantly, the distribution of CSCs in cancerous tissue is not uniform. Research has shown that some glioma stem cells (GSCs) are preferentially located at the invasive front rather than at the core of glioma tissues [12]. There is little research on the relationship between the uniform distribution of CSCs and the stiffness of the matrix. More than 80% of HCC patients have liver cirrhosis or fibrosis. Increased liver stiffness is an important risk factor for the progression of HCC [13]. Both of these observations suggest that the development of HCC is directly related to the stiffness of the matrix. Whether the matrix stiffness affects the CSC phenotype of HCC cells has not been fully elucidated.

To explore the biomechanics related to matrix stiffness, researchers have developed a series of biomaterials. According to previous studies, polyacrylamide (PA) hydrogels are frequently used as synthetic biomaterials for biomechanical research. The swelling properties and tunable stiffness of PA hydrogels can produce hydrogels with mechanical properties similar to those of the extracellular matrix (ECM). The use of a PA hydrogel as a matrix material can accurately simulate mechanical cell stimulation.

In this study, we developed a cell culture model with tunable matrix stiffness to investigate whether matrix stiffness regulates the CSC phenotype of cancer cells.

## 2. Results

### 2.1. Cell Culture on Polyacrylamide Hydrogels with Different Stiffnesses

To explore the effect of matrix stiffness on the stemness of HCC cells, mechanically tunable PA hydrogels were used in this study. The matrix stiffnesses of these hydrogels were altered by different concentrations of methylene-bis-acrylamide. The process for producing the PA hydrogels is shown in Figure 1A. Three hydrogels with different stiffnesses were used to simulate the liver at different stages (5.9 kPa for a normal liver, 24.8 kPa for a fibrotic liver, and 48.1 kPa for cirrhosis and HCC). In this study, “soft” was used to describe a stiffness of 5.9 kPa, “medium” was used to describe a stiffness of 24.8 kPa, and “stiff” was used to describe a stiffness of 48.1 kPa [14]. The PA gels used in this study were coated with collagen I, which is the predominant ECM protein in the liver. Cells of the HCC cell line MHCC97H were then seeded on matrices with different stiffnesses. As the matrix stiffness increased, cell spreading and the cell area increased (Figure 1B,C).

### 2.2. The Matrix Stiffness Regulated Stem Cell-Related Molecular Markers in HCC Cells

Liver cancer cells with stem cell phenotypes generally highly express well-known stem cell-related molecular markers, such as Oct-4, Sox-2, Nanog, Epithelial cell adhesion molecule (EpCAM), CD90, CD133, and CD44 [2]. Thus, qRT-PCR was used to analyze stem cell molecular markers at the mRNA level. Oct-4, Sox-2, CXCR4, CD133, and CD133, five stem cell-related markers, were highly expressed when cells were grown on the soft matrix (Figure 2A). Moreover, CD133+, CD90+ LCSC surface markers, and positive and double-positive cells were more enriched in the soft matrix than in the other matrices, as indicated by flow cytometry (Figure 2D). The low-Hoechst and low-propidium iodide (PI) cell population is referred to as the side population (SP) phenotype and indicates stem cell-like cancer cells in various cancers [15]. We indeed observed that the SP phenotype was retained by a higher proportion of cells grown on the soft matrix than on the other matrices (Figure 2C). These results suggest that cells cultured on the soft matrix exhibited more stem cell-related phenotypes than those grown on the other matrices from a molecular marker perspective. Additionally, there was no significant difference between the medium and stiff matrices.

### 2.3. The Proliferation, Cytoskeleton, Sphere-Forming Ability, Cell Cycle, and Chemoresistance of Cells Grown on Matrices with Different Stiffnesses

To further measure the stemness of MHCC97H cells grown on matrices with different stiffnesses, the Cell Counting Kit-8 (CCK-8) assay was used to detect their proliferation, revealing that increased stiffness enhanced cell proliferation (Figure 3A). Previous studies have shown that CSCs derived from HCC cells are morphologically smaller and rounder than normal HCC cells. In addition, LCSCs have shown less well-defined stress fibers than HCC cells and therefore exhibit a weaker filamentous actin network [16]. Our results showed that the cells grown on the soft matrix had less well-defined stress fibers and a weaker filamentous actin network than those grown on the other matrices (Figure 3B). Next, we assessed the stem cell phenotype of HCC cells based on their intrinsic sphere-forming ability. HCC cells cultured on the soft matrix exhibited more spheres than those cultured on the medium and stiff matrices (Figure 3C). Furthermore, we analyzed the cell cycles of cells grown on the different matrices. We found that cells grown on the soft matrix were mostly in the G0–G1 phase, while fewer cells were in the S and G2–M phases (Figure 3D).

Cancer cells with stemness always exhibit drug or radiation resistance. Our results showed that the cells grown on the soft matrix had higher clonogenic potential than those grown on the medium and stiff matrices. Figure 4 also shows that the cells grown on the soft matrix after sorafenib (10 μM) treatment exhibited strikingly high clonogenic potential (approximately 2.7-fold greater than that of cells grown on the medium matrix and 5.2-fold greater than that of cells grown on the stiff matrix). In addition, cells grown on the soft matrix after Fluorouracil (5-FU) (10 μg/mL) treatment also showed strikingly high clonogenic potential (approximately 3.03-fold greater than that of cells grown on the medium matrix and 4.1-fold greater than that of cells grown on the stiff matrix).

### 2.4. The Matrix Stiffness Regulated the Tumorigenic Potential of HCC Cells In Vivo

Finally, we explored whether the tumorigenic potential of HCC cells changed after the cells were cultured on hydrogels with different matrix stiffnesses. Nude mice were subcutaneously inoculated with HCC cells that were cultured on hydrogels with different matrix stiffnesses. The volumes and weights of the transplanted tumors significantly decreased as the matrix stiffness increased (Figure 5A–C). There was no significant change in body weight between the different groups (Figure 5D). These results further indicate that the matrix stiffness could regulate HCC cell stemness from an in vivo perspective, indicating that the soft matrix could increase HCC stemness.

## 3. Discussion

In this study, we investigated the effects of PA hydrogels on the stemness of HCC cells. Our results showed that the PA hydrogels had great mechanical characteristics and adjustable stiffnesses, and the HCC cells grown on PA hydrogel surfaces coated with collagen I were healthy. Our results showed that HCC cells were poorly spread on the soft matrix and better spread on the stiff matrix. These results are similar to those reported in a previous study [17] and demonstrate that HCC cells can perceive mechanical properties on PA-containing two-dimensional (2D) materials.

CSCs, also known as tumor-initialing cells, are a rare cell subpopulation that has the ability to initiate cancer and induce cancer metastasis [18]. Despite the extensive research on CSCs, there is no consensus on the origin of CSCs. Our results suggest that the matrix stiffness can induce an increase in a series of phenotypic CSC markers in HCC cells. Many factors, including hypoxia, drug treatment, and radiation, have been shown to induce the dedifferentiation of cancer cells to obtain CSC-related phenotypes [19,20,21]. Furthermore, our results demonstrated that biomechanical elements induce cancer cell dedifferentiation both in vitro and in vivo. Our results showed that the soft matrix increased the CSC phenotype of HCC cells, suggesting that less biomechanical stimulation is beneficial for maintaining cell stemness and even for increasing the number of CSCs. HCCs cannot spread normally and show poor proliferation when cancer cells are present in a soft matrix. For cancer cells to survive in a soft matrix, they must express high levels of CSC-related markers to enter a dormant state and adapt to the surrounding environment by inhibiting proliferation or blocking entry into the G0–G1 phase and then initiating tumorigenesis when the microenvironment is suitable. Furthermore, our results may also contribute to further understanding the metastatic mechanism of cancer cells. When cancer cells arrive at some soft sites (e.g., bone marrow), they may be transformed into CSCs due to the soft surroundings, further initiating tumorigenesis at the metastatic site. As the tissue matrix stiffness increases, CSCs are activated and differentiated into more cancer cells to promote the development of cancer.

Interestingly, our results showed that there was no significant change in the phenotype of CSCs when the matrix stiffness increased from medium to stiff. Previous studies have also shown that cells have a range of mechanical stimulation perceptions and that cells spread well in a certain range, which includes stiffnesses that are not too small or too large [22]. Furthermore, our results also demonstrated that cellular perceptual mechanical stimuli function in a certain range. All the biomechanical effectors may be activated by a hydrogel with medium stiffness. A continuous increase in the stiffness does not further enable cells to react. However, a certain number of CSCs is needed in cancer tissues to promote cancer progression. When tissue is soft, more CSCs are enriched, and when the stiffness increases, a certain number of CSCs differentiate into cancer cells, thus decreasing the number of CSCs. When the stiffness continues to increase, CSCs may continue to differentiate, but a minimum cell number is maintained, thus maintaining the progression of the whole tumor.

Our study explored the relationship between mechanical factors and HCC cell stemness using PA hydrogels. The results showed that a soft matrix stiffness increases HCC cell stemness. Our research showed that PA can be applied in cancer research and may provide experimental evidence for research on cancer progression. Our study is the first to evaluate the effect of matrix stiffness on the phenotype of HCC stem cells with multiple indicators. However, we did not further explore the underlying molecular mechanism, and further research is thus needed.

## 4. Materials and Methods

### 4.1. Preparation of Polyacrylamide Hydrogels

PA hydrogels with different stiffnesses were prepared to simulate certain stages of the liver matrix. Forty percent acrylamide (Sigma-Aldrich, Saint Louis, MO, USA), 2% methylene-bis-acrylamide (Sigma-Aldrich) and distilled water were combined according to the different proportions of PA solution desired, and PA gel polymerization was promoted by the addition of 10% ammonium persulfate (APS; 1/200 volume) and tetramethylethylenediamine (TEMED; 1/2000 volume). A solidified PA hydrogel was then removed carefully and stored in deionized water overnight for analysis. An adequate volume of 0.2 mg/mL sulfosuccinimidyl 6-(4′-azido-2′-nitrophenylamino) hexanoate (sulfo-SANPAH; Thermo Fisher Scientific, Waltham, MA, USA) solution (crosslinking agent) was added to the gel surface and then irradiated with ultraviolet (UV) light for 25 min. After complete washing with phosphate-buffered saline (PBS) and Hepes buffer, an adequate volume of 0.2 mg/mL collagen I (Shengyou Biotechnology Co., Hangzhou, China) was added and crosslinked securely onto the PA hydrogel surface [23]. To measure the mechanical stiffness of the hydrogels, planar PA hydrogels with various bis-acrylamide concentrations were cut into rectangular strips (length:width:height = 5:2:0.1 cm). The strain, defined as the change in the length (DL)/the initial length (L), was measured using a self-weighing assay, which yielded Young’s modulus using the following equation: E = (G/A)/(DL/L), where G and A are the weight and cross-sectional area, respectively [24]. The stiffnesses of the hydrogels with different proportions are shown in Table 1.

### 4.2. Cell Culture

The human hepatoma cell line MHCC97H was obtained from the Liver Cancer Institute, Zhongshan Hospital, Fudan University (Shanghai, China) and maintained as a monolayer culture in high-glucose Dulbecco’s Modified Eagle’s Medium (H-DMEM; Gibco, Pleasanton, CA, USA) supplemented with 10% fetal bovine serum (FBS; HyClone, Logan, UT, USA), penicillin (100 U/mL), and streptomycin (100 U/mL) in an incubator at 37 °C with 5% CO_2_. MHCC97H cells were seeded on the PA hydrogels for 48 h for subsequent experiments.

### 4.3. Real-Time qRT-PCR

Total RNA was extracted from the cells using TRIzol reagent (TaKaRa, Kyoto, Japan), and equal amounts of RNA were reverse-transcribed into cDNA using Prime Script RT Master Mix (TaKaRa, Kyoto, Japan). Real-time quantitative reverse transcription-polymerase chain reaction (qRT-PCR) was performed using a CFX96™ real-time PCR detection system (Bio-Rad, Hercules, CA, USA) with SYBR Green Master Mix (TaKaRa) according to the manufacturer’s instructions. The results were analyzed using CFX Manager Software software (Bio-Rad). The following specific primers were used for the amplification of several genes: β-actin: sense 5′ AAAGACCTGTACGCCAACAC 3′, antisense 5′ GTCATACTCCTGCTTGCTGAT 3′; Oct4: sense 5′ GCAGCGACTATGCACAACGA 3′, antisense 5′ AGCCCAGAGTGGTGACGGA 3′; Sox2: sense 5′ ATGCACCGCTACGACGTGAG 3′, antisense 5′ GCCCTGGAGTGGGAGGAAGA 3′; CXCR4: sense 5′ AGGAAATGGGCTCAGGGG 3′, antisense 5′ AGGAAATGGGCTCAGGGG 3′; CD90: sense 5′ AGCCTTCGTCTGGACTGCC 3′, antisense 5′ TGGTTCGGGAGCGGTATGT 3′; and CD133: sense 5′ TGGAGCGTCCCTTCACCC 3′, antisense 5′ TTTCTCAAAGTATCTGGATGTAGCA 3′.

### 4.4. Flow Cytometry

The following anti-human monoclonal antibodies (mAbs) were used for flow cytometry. A phycoerythrin (PE)-conjugated mouse IgG1 isotype control antibody (Biolegend, San Diego, CA, USA) and an allophycocyanin (APC)-conjugated mouse IgG1 K isotype control antibody (Biolegend) were used as isotype controls. A PE-conjugated anti-human CD133 antibody and an APC-conjugated anti-human CD90 (Biolegend) antibody were used for surface marker analysis. Flow cytometry was performed using a FACS Canto II (BD, San Diego, CA, USA) flow cytometer equipped with software. Side scatter and forward scatter profiles were used to eliminate cell doublets. The results are presented as the percentage of positive cells.

Cells were harvested from the PA hydrogels and dissociated into single cells. After washing the cells with PBS, the cells were fixed using ice-cold 70% ethanol and stored at 4 °C overnight. The cells were then stained using a cell cycle kit (BD) according to the manufacturer’s instructions and analyzed on a BD FACS Canto II (BD) flow cytometer.

### 4.5. Hoechst Dye Exclusion Assay

Cells were harvested from the PA hydrogels and dissociated into single cells. A total of 5 μg of Hoechst 33342 dye (Thermo Fisher Scientific) was added per 1 × 10^6^ cells, which were incubated at 37 °C in the dark with intermittent shaking for 90 min. The ATP-binding cassette (ABC) transporter inhibitor verapamil (50 μM) (Sigma-Aldrich) was used to block Hoechst efflux to set the “low-Hoechst” gate. The stained cells were washed and resuspended in ice-cold PBS containing 2 mg/mL PI. Flow cytometry was performed on a BD FACS Canto II flow cytometer with a UV excitation wavelength of 355 nm and emission wavelengths of 450/50 nm (Hoechst blue) and 675/50 nm (Hoechst red). The gating strategy ensured that only live (PI-excluded) and single (doublet-excluded) cells were analyzed. For the remaining cells, the least fluorescent population was termed the SP, and the percentage was determined [25].

### 4.6. Proliferation Assay

A CCK-8 assay was employed to evaluate the proliferation of MHCC97H cells at different matrix stiffnesses. The cells were cultured on the PA hydrogels for 48 h. The culture medium was removed from the 24-well plate, and 1000 mL of serum-free medium was added to each well. The cultures were then treated with 100 μL of the CCK-8 reagent (Sigma-Aldrich) for 2 h. Following this color change, 100 mL of liquid was sampled and loaded onto a 96-well plate. After a 2-min oscillation, optical density (OD) values were evaluated at 450 nm using a microplate reader (Bio-Rad).

### 4.7. Clonogenic Assay

After culturing the cells on PA hydrogels at various stiffnesses for 48 h, sorafenib (10 μM), 5-FU (10 μg/mL) or nothing was added to the medium. The cells were then retrieved by trypsinization, counted and plated at a clonal density (1000 cells/well) in 6-well plates in normal medium. Next, the colonies were stained with 0.1% crystal violet for 30 min and washed with PBS to remove the excess stain, and images were then acquired.

### 4.8. Sphere Formation Assay

Cells were harvested from the PA hydrogels and dissociated into single cells. Next, 1000 cells were added to each well of 24-well culture plates containing a specific medium for tumor sphere cells that included DMEM/F-12 supplemented with B27 without vitamin A, 1% N-2, 20 ng/mL epidermal growth factor (PeproTech, Rocky Hill, NJ, USA), and 10 ng/mL basic fibroblast growth factor (Life Technologies, Pleasanton, CA, USA). Then, the cells were cultured for an additional 7–10 days in the medium until the sphere diameter was greater than 50 μm. The spheres in the suspension cultures were counted on a widefield light microscope.

### 4.9. Immunostaining

Cells were cultured on the PA hydrogels for 48 h. Then, the cells were fixed with 4% formaldehyde in PBS for 30 min in the dark and permeabilized with 0.25% Triton X-100 and 2% bovine serum albumin (BSA) in PBS for 15 min. After saturating the cells with 5% bovine serum, the cells were incubated with tetramethylrhodamine (TRITC)-phalloidin (Sigma-Aldrich) at 4 °C overnight and then photographed with a confocal microscope (Leica, Solms, Germany).

### 4.10. Evaluation of Tumorigenesis by Subcutaneous Xenotransplantation into Nude Mice

To evaluate whether the tumorigenesis of HCC cells changes after culture on hydrogels with various stiffnesses, animal experiments were performed. Fifteen 4-week-old female nude mice were purchased from the Laboratory Animal Center, Chongqing Medical University, China, and randomly divided into three groups. In addition, cells were harvested from the PA hydrogels and dissociated into single cells. Approximately 2 × 10^6^ cells in 0.2 mL of medium with 50% Matrigel (BD) were subcutaneously injected into the mice followed by 3 weeks of standard feeding. The tumor sizes (length and width) and weights were measured with a caliper once every other day until the mice were sacrificed. The tumor volume was calculated according to the following previously described equation: V = 1/2 (L × W^2^) [6]. The animal experiments were performed in triplicate.

### 4.11. Statistical Analysis

All data are presented as the mean ± standard deviation. The data from each group were analyzed by one-way analysis of variance (ANOVA) followed by a *t*-test to analyze significant differences relative to the control group. A *p*-value of *p* < 0.05 was considered statistically significant.

## Figures and Tables

**Figure 1 ijms-20-02831-f001:**
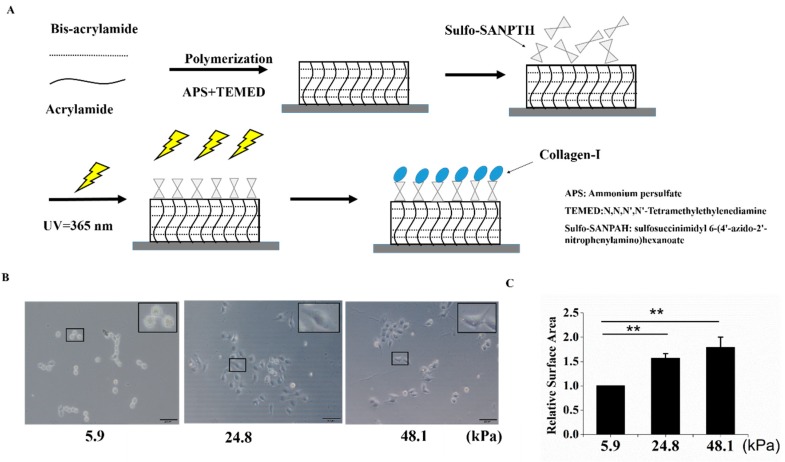
A schematic of polyacrylamide (PA) hydrogel preparation and cell seeding. (**A**) A schematic of PA hydrogel crosslinking and coating. (**B**,**C**) The stiff matrix induced cell spreading, and relative cell spreading area statistics are shown. Bar = 200 μm. The values are presented as the mean ± standard deviation (SD) of three independent experiments. *n* = 3; and ** *p* < 0.01.

**Figure 2 ijms-20-02831-f002:**
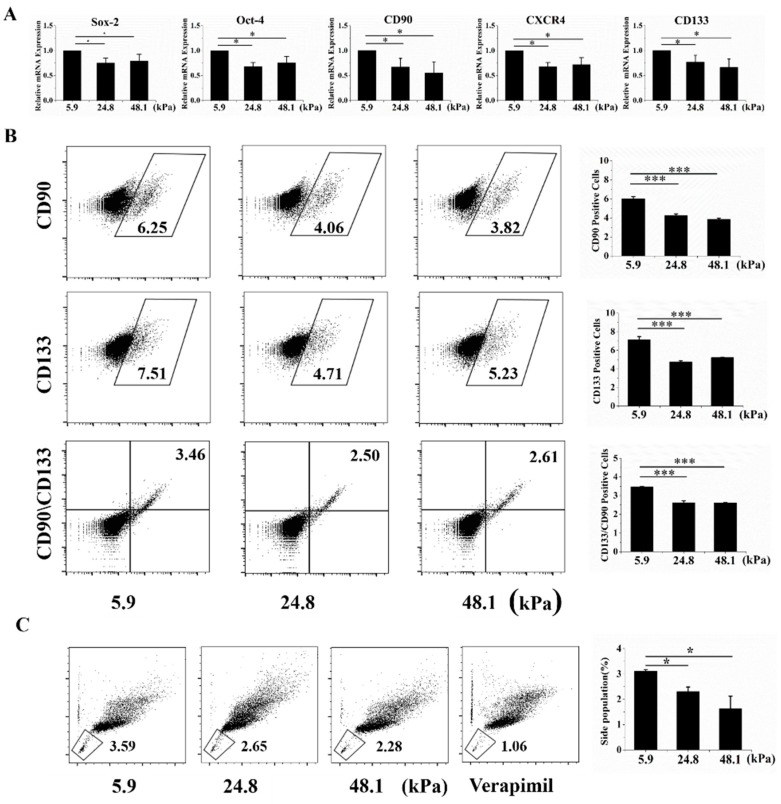
The soft matrix enhanced the expression of molecular markers of stemness. (**A**) The soft matrix increased the mRNA expression levels of the cancer stem cell markers Oct-4, Sox-2, CXCR4, CD133, and CD90, as indicated by qRT-PCR. All the results are normalized to 5.9 kPa. (**B**) Flow cytometry showed that the soft matrix increased the proportion of CD90+, CD133+, and CD90+CD133+ cells. (**C**) The soft matrix increased the number of side population (SP) cells, and the cells were stained with a negative control. The values are presented as the means ± SD of three independent experiments. *n* = 3; * *p* < 0.05; *** *p* < 0.001.

**Figure 3 ijms-20-02831-f003:**
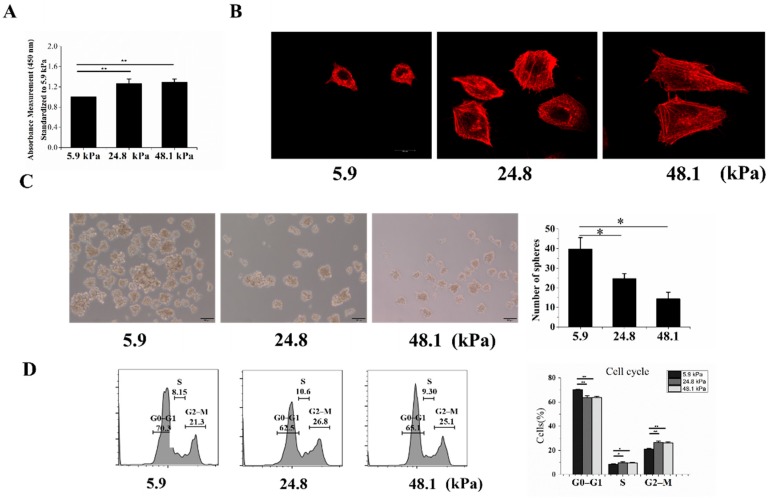
The proliferation, cytoskeletons, sphere formation abilities, cell cycles, and chemoresistance of cells grown on matrices with different stiffnesses. (**A**) The proliferation of MHCC97H cells was detected by the CCK-8 assay. (**B**) F-actin in MHCC97H cells was stained by phalloidin and imaged by confocal microscopy. (**C**) The soft matrix increased the sphere-forming ability, and the results were statistically significant. (**D**) Graphs showing the cell cycle analysis and % of the population in each phase. Bar = 20 μm (**B**), Bar = 100 μm (**C**). The values are presented as the mean ± SD of three independent experiments. *n* = 3; * *p* < 0.05; ** *p* < 0.01.

**Figure 4 ijms-20-02831-f004:**
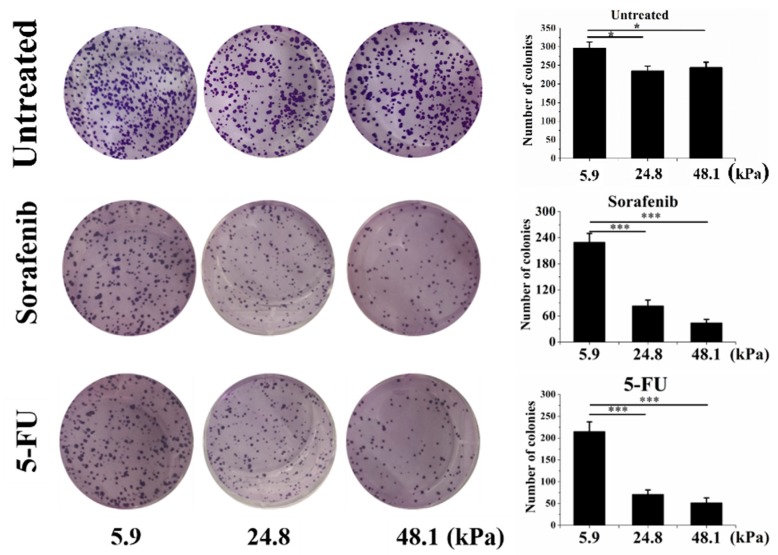
The chemoresistance of cells grown on matrices with different stiffnesses. The untreated control (top), sorafenib (10 μM; middle) and Fluorouracil (5-FU) (10 μg/mL; bottom) groups are shown. The values are presented as the mean ± SD of three independent experiments. *n* = 3; * *p* < 0.05; *** *p* < 0.001.

**Figure 5 ijms-20-02831-f005:**
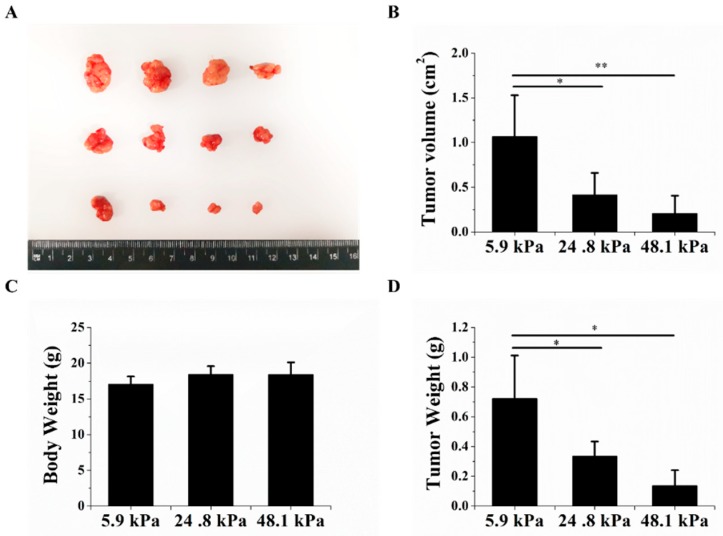
The matrix stiffness regulated tumorigenic potential. (**A**) The soft matrix increased tumor formation; soft matrix (top), medium matrix (middle), and stiff matrix (bottom). (**B**,**C**) The soft matrix increased the subcutaneous xenograft weight and volume. (**D**) Changes in the body weights of nude mice 23 days after injection. The values are presented as the mean ± SD of three independent experiments. *n* = 4; * *p* < 0.05; and ** *p* < 0.01.

**Table 1 ijms-20-02831-t001:** Expected modulus of elasticity values after polymerization of relative acrylamide and bis-acrylamide concentrations.

Acrylamide (%)	Bis (%)	40% Acrylamide (mL)	2% Bis (mL)	H_2_O (mL)	E (kPa)
10	0.03	2.5	0.15	7.35	5.9 ± 2.227854
10	0.10	2.5	0.50	7.00	15.1 ± 1.563117
10	0.13	2.5	0.65	6.85	24.8 ± 2.821938
10	0.25	2.5	1.25	6.25	43.4 ± 9.508067
10	0.26	2.5	1.30	6.20	48.1 ± 9.798639
10	0.30	2.5	1.50	6.00	53.4 ± 4.119871
10	0.35	2.5	1.75	5.75	60.8 ± 8.555992
10	0.45	2.5	2.25	5.25	81.7 ± 3.557152

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
