# Peer review of "A Soft Matrix Enhances the Cancer Stem Cell Phenotype of HCC Cells"

_ijms, 2019, doi:10.3390/ijms20112831_

Round 1
Reviewer 1 Report
Manusript titled " A soft matrix enhances the cancer stem cell phenotype of HCC cells" describes the role soft matrix in inducing stemness in HCC cells in vitro and on tumorigenic potential of HCC cells. Overall this is an interesting paper.
Authors needs to show what signaling pathways gets activated in the matrix that induces stemcellness. Does the tumor formed in vivo still consists of cancer stem cells? How matirx affects cell proliferation in addition to inducing stem cell features? If authors can show BrdU assay or PCNA/Ki67 staining to show cell proliferation will be good.
Author Response
Responses to Reviewer 1 Comments
English language and style
(x) Moderate English changes required
Response: Thank you for your careful revisions. The English grammar and syntax in this manuscript have been revised by the AJE company.
Point 1: Authors needs to show what signaling pathways gets activated in the matrix that induces stemcellness.
Response 1: We are grateful for your helpful advice. The next plan in our experimental design is to detect the relevant signaling pathways. We are preparing for the signaling pathways study, but there are no corresponding results to report yet. An extensive study of the signaling pathways is under investigation in our laboratory, and we expect that the results will form a complete story of their own.
Point 2: Does the tumor formed in vivo still consists of cancer stem cells?
Response 2: Thank you for your interest in this topic. Cancer stem cells exist in most cancer tissues, although the number of cancer stem cells may be small. Zhu et al.[1] showed that a small proportion of mouse xenomas formed by hepatocellular carcinoma cells still contain cancer stem cell markers. We speculate that the tumors formed in vivo in this study consisted of some cancer stem cells.
[1] Zhu, Z., X. Hao, M. Yan, M. Yao, C. Ge, J. Gu, and J. Li. "Cancer Stem/Progenitor Cells Are Highly Enriched in Cd133+Cd44+ Population in Hepatocellular Carcinoma." Int J Cancer 126, no. 9 (2010): 2067-78.
Point 3: How matrix affects cell proliferation in addition to inducing stem cell features? If authors can show BrdU assay or PCNA/Ki67 staining to show cell proliferation will be good.
Response 3: Thank you for your kind suggestion. In our study, we used the CCK-8 assay to detect how the matrix affects cell proliferation (Figure R). Our results show that increased matrix stiffness enhances cell proliferation, while a softer matrix stiffness inhibits cell proliferation. This may be one of the reasons why soft substrates can maintain cell stemness. Decreased proliferative capacity may allow cells to enter a quiescent state. According to your suggestion, we added the results of cell proliferation in Figure 1 3A.
Figure R. Matrix stiffness affects MHCC97H cell proliferation. The values are presented as the mean ± SD of three independent experiments. n = 3; **p < 0.01.
Reviewer 2 Report
In this manuscript submitted by Tian et al., the authors described the use of mechanically tunable PA hydrogels to stimulate different stiffness of the liver tissue matrix and demonstrated that soft matrix increased the expression of liver cancer stem cell markers and the number of SP cells. In addition, the authors also reported the increased in early cell cycle arrest at G1 phase and sphere forming ability with concomitant increased in chemoresistance and tumorigenic potential. The results presented are interesting, however suffers several major weakness that need to be addressed.
Major comments:
1. Liver cancer stem cells or LCSCs are still controversial and has not be definitively demonstrated to exist and their origin(s) are still unclear. Hence, the authors might consider to use liver cancer cells with stemness properties to describe these cells instead.
2. Did the authors tried to compare “CSCs” versus their non-“CSCs” counterparts in terms of stemness properties, sphere-forming ability, cell cycle and chemoresistant profiles? This is important as such properties should be unique to these “CSCs” and not the tumor bulk cells.
3. EpCAM is one of the first and notable liver “CSC” marker described by Yamashita et al. in 2009. The authors should test this liver-related CSC marker in their study.
4. Could the authors show the corresponding untreated control for Fig 3D?
5. Fig 4 is interesting, however, in order to relate these changes to stiffness affecting “CSC” properties, the authors should show what are the changes in CSC markers expression in these different groups.
6. Pages 156/157; “Our results may provide evidence for the origin of CSCs because CSCs may arise from normal cancer cells.” This statement is not well-demonstrated and should be removed. The origin of liver CSCs is still controversial.
7. Pages 166 to 168; The statement: “our results suggested that a softer matrix induces the transformation of cancer cells into CSCs, indicating that CSCs may appear in the early stage of cancer onset but in a dormant state.” This is not well-demonstrated or supported by the results presented. I suggests the authors to remove this sentence.
Minor comments:
1. Page 1, line 27; “Liver cancer stem cells (LCSCs) compose a small portion of HCC cells[1].” This is a wrong citation.
2. Figure annotation for Fig 1C is missing.
3. What is the statistical method used for Fig 3C. Please explain.
Author Response
Responses to Reviewer 2 Comments
English language and style
(x) Extensive editing of English language and style required
Response: Thank you for your careful revisions. The English grammar and syntax in this manuscript have been revised by the AJE company.
Major Comments
Point 1: Liver cancer stem cells or LCSCs are still controversial and has not be definitively demonstrated to exist and their origin(s) are still unclear. Hence, the authors might consider to use liver cancer cells with stemness properties to describe these cells instead.
Response 1: We are grateful for your helpful advice. We have changed the description of the cells concerned according to your suggestion.
Point 2: Did the authors tried to compare “CSCs” versus their non-“CSCs” counterparts in terms of stemness properties, sphere-forming ability, cell cycle and chemoresistant profiles? This is important as such properties should be unique to these “CSCs” and not the tumor bulk cells.
Response 2: Thank you for your kind suggestion. Our previous studies compared the differences between CSCs and non-CSCs (Sun et al.)[1]. We enriched CSCs from the MHCC97H cell line and compared the differences between CSC and non-CSC counterparts in terms of stemness properties, sphere formation ability and chemoresistant profiles. Our previous results showed that CSCs have stronger stemness properties, sphere formation abilities and chemoresistant profiles than tumor bulk cells.
[1] Sun, J., Q. Luo, L. Liu, B. Zhang, Y. Shi, Y. Ju, and G. Song. "Biomechanical Profile of Cancer Stem-Like Cells Derived from Mhcc97h Cell Lines." J Biomech 49, no. 1 (2016): 45-52.
Point 3: EpCAM is one of the first and notable liver “CSC” marker described by Yamashita et al. in 2009. The authors should test this liver-related CSC marker in their study.
Response 3: Thank you for your careful revisions. According to previous reports, many markers can be used to characterize liver cancer stem cells, such as CD133, CD90, CD44, Oct-4, Sox-2, Nanog, EpCAM, CXCR4, and OV-6[2-4]. According to previous reports, we selected CD133, CD90, Oct-4, Sox-2, and CXCR4 as markers to characterize liver “CSCs”. We completely agree with the point you addressed. We will utilize EpCAM as a liver “CSC” marker and test it in our next study.
[2] Song, Y., G. Pan, L. Chen, S. Ma, T. Zeng, T. H. Man Chan, L. Li, Q. Lian, R. Chow, X. Cai, Y. Li, Y. Li, M. Liu, Y. Li, Y. Zhu, N. Wong, Y. F. Yuan, D. Pei, and X. Y. Guan. "Loss of Atoh8 Increases Stem Cell Features of Hepatocellular Carcinoma Cells." Gastroenterology 149, no. 4 (2015): 1068-81 e5.
[3] Han, S., J. Guo, Y. Liu, Z. Zhang, Q. He, P. Li, M. Zhang, H. Sun, R. Li, Y. Li, W. Zeng, J. Liu, L. Lian, Y. Gao, and L. Shen. "Knock out Cd44 in Reprogrammed Liver Cancer Cell C3a Increases Cscs Stemness and Promotes Differentiation." Oncotarget 6, no. 42 (2015): 44452-65.
[4] Schrader, J., T. T. Gordon-Walker, R. L. Aucott, M. van Deemter, A. Quaas, S. Walsh, D. Benten, S. J. Forbes, R. G. Wells, and J. P. Iredale. "Matrix Stiffness Modulates Proliferation, Chemotherapeutic Response, and Dormancy in Hepatocellular Carcinoma Cells." Hepatology 53, no. 4 (2011): 1192-205.
Point 4: Could the authors show the corresponding untreated control for Fig 3D?
Response 4: We appreciate these suggestions. Figure R is the result of the untreated control group. According to your suggestion, we have added this result and related descriptions (including figure annotation and experimental methods) in the revised manuscript.
Given this addition, to make the figures clearer, we split Figure 3 into two new figures in the revised manuscript (Figure 3 and Figure 4).
Figure R. Clonogenic potential is affected by matrix stiffness without drug treatment. The values are presented as the mean ± SD of three independent experiments. n = 3; *p < 0.05.
Point 5: Fig 4 is interesting, however, in order to relate these changes to stiffness affecting “CSC” properties, the authors should show what are the changes in CSC markers expression in these different groups
Response 5: Thank you for your interests. Our main objective is to demonstrate whether matrix stiffness affects the stemness of hepatocellular carcinoma cells. Tumorigenesis experiments in vivo have been recognized as a gold standard in cancer stem cell research. In the present study, we aimed to determine whether the stemness of cells at different matrix stiffnesses differred by comparing tumor sizes. Whether the internal composition of tumors is different after tumorigenesis was not the focus of our attention; therefore, we cannot provide the results of changes in CSC marker expression in the different groups in this study.
We thank you again for your constructive comments and will evaluate the changes in CSC marker expression in these different groups in the future.
Point 6: Pages 156/157; “Our results may provide evidence for the origin of CSCs because CSCs may arise from normal cancer cells.” This statement is not well-demonstrated and should be removed. The origin of liver CSCs is still controversial
Response 6: We are grateful for your helpful advice. According to your suggestion, we deleted this statement in the revised manuscript.
Point 7: Pages 166 to 168; The statement: “our results suggested that a softer matrix induces the transformation of cancer cells into CSCs, indicating that CSCs may appear in the early stage of cancer onset but in a dormant state.” This is not well-demonstrated or supported by the results presented. I suggest the authors to remove this sentence.
Response 7: Thank you for your kind suggestion. According to your suggestion, we have removed this sentence.
Minor comments:
Point 1: Page 1, line 27; “Liver cancer stem cells (LCSCs) compose a small portion of HCC cells [1].” This is a wrong citation.
Response 1: Thank you for your helpful correction. We have changed the reference location.
Point 2: Figure annotation for Fig 1C is missing.
Response 2: Thank you for your helpful correction. We have added the corresponding Figure annotation.
Point 3: What is the statistical method used for Fig 3C. Please explain.
Response 3: Thank you for your interest in the data statistics. The original data were analyzed by FlowJo software to set the gate. The % population in each phase was statistically significant, and a t-test was used to analyze significant differences. We determined whether the cell cycle changed by comparing the percentage of cells in the same period between different experimental groups.
Round 2
Reviewer 2 Report
The authors have addressed each of the points raised in my initial review. I have no further comments.